# Record-breaking Greenland ice sheet melt events under recent and future climate

Josep Bonsoms [1] ✉, Sergi González-Herrero [2], Xavier Fettweis [3], Marc Lemus-Cánovas [4], Marc Oliva[1] & Juan I. López-Moreno[5]

The Greenland Ice Sheet (GrIS) has experienced a strong intensification of summer surface melting, with extreme events becoming more frequent, extensive, and severe. Despite its importance for global sea-level rise, the mechanisms driving these extremes remain incompletely understood. We analyze extreme melting events over 1950–2023 using an analog-based framework combined with a regional climate model to disentangle thermodynamic and dynamic contributions. Thermodynamic processes intensify meltwater production by 25% relative to 1950–1975 when circulation analog events are included, increasing to 63% when all extreme melting events are included, with the strongest increases in northern Greenland. Seven of the ten most extreme events occurred after 2000, with meltwater anomalies reaching up to three times their synoptic average. Record-breaking events such as August 2012, July 2019, and July 2021 show no dynamic precedents. Future projections under high-emission scenarios suggest that extreme meltwater anomalies could increase by up to +372% by 2100 (SSP5-8.5, CMIP6), highlighting the profound impact of climate change on GrIS melt extremes.

The Greenland Ice Sheet (GrIS), holding an ice volume of $2.96 \times 10^6$ km³ (equivalent to 7.4 m of global sea-level rise)[1] has undergone accelerating mass loss, averaging $169 \pm 9$ Gt yr⁻¹ (1992–2020) predominantly from surface processes[2]. This loss has intensified since the 1990s[3,4], and contributed $3.5 \pm 0.2$ mm yr⁻¹ to sea-level rise (2005–2017)—nearly equivalent to the combined input from all glaciers globally[5]. Additionally, extreme melting events (see "Methods" section) have increased in frequency and magnitude, particularly in north-western (NW) and northern (NO) sectors[6], with events like 2012 and 2019 exceeding both satellite-era records[7,8] and palaeoclimate limits from the Holocene[9]. The 2019 season saw melting at the ice-sheet summit[10] and total losses of 444 Gt yr⁻¹ (167% above the 1992–2020 mean)[2].

Atmospheric circulation has long been identified as a key driver of GrIS melt variability[11–14]. High-pressure systems over or near Greenland alter poleward moisture transport and cloud cover, thereby affecting the surface energy balance that governs melting[15,16]. Increasing

blocking events enhanced surface melting, contributing to the rise in extreme runoff recurrence in GrIS[17]. Thermodynamic processes, such as radiative forcing, surface albedo decline and cloud microphysical feedback, have also been linked to extreme melting records[18,19]. Although previous analyses have established links between extreme melt events and atmospheric circulation patterns, the extent to which meltwater production has increased under similar anticyclonic and omega-blocking patterns remains unclear. Clarifying this distinction is key to disentangling the role of large-scale atmospheric dynamics from surface-driven changes (e.g., warmer air temperatures, enhanced radiative forcing, and the expansion of meltwater-prone areas). Extreme melting events are important to analyze as they hold a contribution of around 30-40% of the GrIS melt [6]. Moreover, they can trigger cascading impacts both on and beyond the ice-sheet, including positive melt-albedo feedbacks[20,21], reduced firn permeability and accelerated ice dynamics[22], widespread ice slab expansion[23], enhanced meltwater delivery to the ice-sheet bed via crevasses, moulins[24].

[1]Department of Geography, Universitat de Barcelona, Barcelona, Spain. [2]WSL Institute for Snow and Avalanche Research SLF, Davos, Switzerland. [3]Laboratory of Climatology, Department of Geography, SPHERES Research Unit, University of Liège, Liège, Belgium. [4]Eurac Resarch, Bolzano, Italy. [5]Instituto Pirenaico de Ecología, Consejo Superior de Investigaciones Científicas (IPE–CSIC), Zaragoza, Spain. ✉e-mail: josepbonsoms5@ub.edu

Extreme melt seasons form persistent low-permeability ice layers, promoting inland expansion of ice slabs and firn aquifers, altering near-surface hydrology, and affecting the ice sheet's response to subsequent melting beyond a single season[22]. An increase in the amount of meltwater produced that leaves the ice sheet (runoff) could lead to large-scale oceanographic disruptions from freshwater discharge, affecting the Atlantic Meridional Overturning Circulation (AMOC), the Earth's climate-system[25,26] and mid-latitude weather systems over Europe[27].

Understanding extreme melt events over the GrIS requires disentangling the roles of dynamical and thermodynamical factors. While anthropogenic climate change can affect both components, its influence on dynamical drivers is harder to detect due to the large natural variability in atmospheric flows[28,29]. Separating these contributions is therefore crucial to accurately understand recent increases in extreme climate events within a climate change context, addressing a knowledge gap noted by the IPCC AR6[29]. Furthermore, projections of future extreme meltwater remain limited, leaving a critical aspect of GrIS mass loss poorly constrained and with still high uncertainty[30]. To our knowledge, no study has yet quantified the partitioning of thermodynamic and circulation contributions to meltwater production during extreme events in the GrIS, nor assessed the future intensification of such extreme melt.

This work analyzes recent changes in the spatial extent and magnitude of extreme melting events (1950-2023) across the GrIS, using satellite observations and meltwater output from a high-resolution regional climate model validated against in situ data and other models[31,32]. We disentangle the thermodynamic and dynamic contributions of 10 most extreme GrIS melting events using an analog-based framework[29,30,33–37]. Furthermore, we assess how summer (June, July and August) extreme melting magnitudes are projected to evolve under high-emission scenarios (RCP8.5 and SSP5-8.5) from CMIP5 and CMIP6, identifying regional hotspots of intensification toward the end of the 21st century.

## Results
### Extreme melting trends and events classification
Extreme melting events over the GrIS have become statistically significant ($p \leq 0.05$) more intense and widespread since the 1990s (Fig. 1). Compared to 1950–1975, simulations show up to eight additional extreme melting days per year between 2000 and 2023 (Fig. 1a), consistent with GrIS extreme melting frequency trends[6]. From 1950 to 2023, the trend in extreme melting magnitude increased sixfold, rising from 12.7 Gt per decade (detrended, −D) to 82.4 Gt per decade after 1990 ($p = 0.01$). Over the same period, the spatial extent of extreme melting expanded by 2.8 million km² per decade (p=0.05). Fig. 1b summarizes the top 10 extreme events (see "Methods" section), including peak daily meltwater, event duration, and total meltwater. These events occurred during July and August, with most of them (7 out of 10), including those of greatest magnitude and duration, taking place between 2000 and 2023. The highest peak daily meltwater was simulated for the August 2019 and July 2012 events, reaching 17 Gt/day (detrended, −D). Total accumulated meltwater per event peaked during the July and August 2012 events, each surpassing 250Gt (−D). Detailed statistics for the top 10 extreme melting events are provided in Table S1.

### Thermodynamic and dynamic attribution
A synoptic classification (see "Methods" section) was performed to assign each day to its corresponding circulation-weather type (CWT) during July and August for the period 1950–2023. The analysis distinguishes between CWTs associated with lower sea-level pressure (SLP) (CWTs 1–12), indicative of cyclonic systems, and those associated with higher SLP (CWTs 13–20), representing mostly anticyclonic weather systems (Fig. S1). The top 10 extreme melting events are predominantly associated with anticyclonic CWTs (13–20) (Table S1). This classification enables the quantification of the increase in meltwater over time for the same anticyclonic CWTs. Specifically, the average meltwater associated with CWTs 13–20 has increased during the 2000–2023 period (Fig. 2c). Details of the meltwater increase for each CWT are shown in Figs. S2 and S3. Among the top 10 extreme events that occurred during the 2000–2023 period (7 events), the average CWT-specific anomaly in meltwater production increased by a factor of three ( + 305%). Event peaks range from +427% (July 2019) to +172% (August 2021) relative to the CWT-specific average during the 1950–1975 period, which was only marginally affected by climate change (Fig. 2).

The increase in sea-level pressure (SLP) during July and August aligns with recent trends in 500 hPa geopotential height (Z500) and 850 hPa air temperature (T850) over the period 1990–2023 (Fig. S4). Z500 and T850 show statistically significant increases over the past few decades and exhibit a strong linear correlation ($r = 0.85, p < 0.001$), indicating enhanced thermal expansion of Z500.

To isolate the role of atmospheric circulation and thermodynamics in driving extreme melting events, a flow analog method (see "Methods" section) is applied using SLP data (Fig. 3). A sensitivity analysis was conducted across different analog flow set sizes (10, 20, 30), time periods, and SLP non-detrended (raw, -R) and detrended (-D), identifying analogs within a 20-day window around the extreme melting events. SLP anomalies for the top 10 extreme melting events, relative to the 1950–1975 baseline, reveal that omega-blocking configurations were dominant during events such as July 2012 and 2019, with SLP anomalies exceeding +10 hPa.

Analogs based on SLP-R reveal a marked increase in the frequency of high-pressure systems over the 1950–2023 period (Fig. 3 and Fig. S5). To isolate this trend, SLP was detrended (SLP-D), enabling a more consistent analog frequency across time and removing the influence of long-term high-pressure increases on the analog selection. The evolution of analog magnitude and frequency based on SLP-D (Fig. 3b, c) confirms that detrending enables the analog search to capture circulation patterns independently of long-term background trends. Notably, the absence of analogs for several of the top ten extreme meltwater events (e.g., August 2012, July 2019, and July 2021) highlights the exceptional dynamical conditions underlying these events (Fig. 3c and Table S2).

The evolution of meltwater production associated with analog events (SLP-D and 10 analogs), along with their temporal trends and cumulative frequency distributions, reveals a thermodynamic increase across the GrIS (Fig. 4). Meltwater during extreme events with historical analogs from the 1950–1975 period increased by 25% relative to 1950–1975, with the largest anomaly recorded in August 2023 ( + 34%) (Table S2). Including all top 10 extreme melt events compared to their analogs, the increase corresponds to a 63% intensification. This meltwater production amplification is particularly pronounced in the north-eastern (NE) ( + 78%) and northern ( + 75%) sectors of the GrIS (Fig. 4b).

### Future extreme melting intensification
The projected upper bounds of extreme melting under future climate scenarios are further analyzed by comparing July–August 95th percentile melt thresholds from the IPCC AR6 baseline period (1986–2005) with those from the 2000–2023 reanalysis (MAR-ERA5) and late-century (2090–2100) projections from MAR-CMIP5 and MAR-CMIP6 under high-end emissions scenarios (RCP8.5 and SSP5-8.5, respectively). Results indicate that meltwater during 2000–2023 (MAR-ERA5) has already increased by 14% relative to the IPCC AR6 baseline climate. Future projections show a marked rightward shift in the probability density function of meltwater production, particularly in CMIP6 simulations (Fig. 5). Projected increases in extreme melting reach +234% in MAR-CMIP5 and +372% in MAR-CMIP6 (multi-model

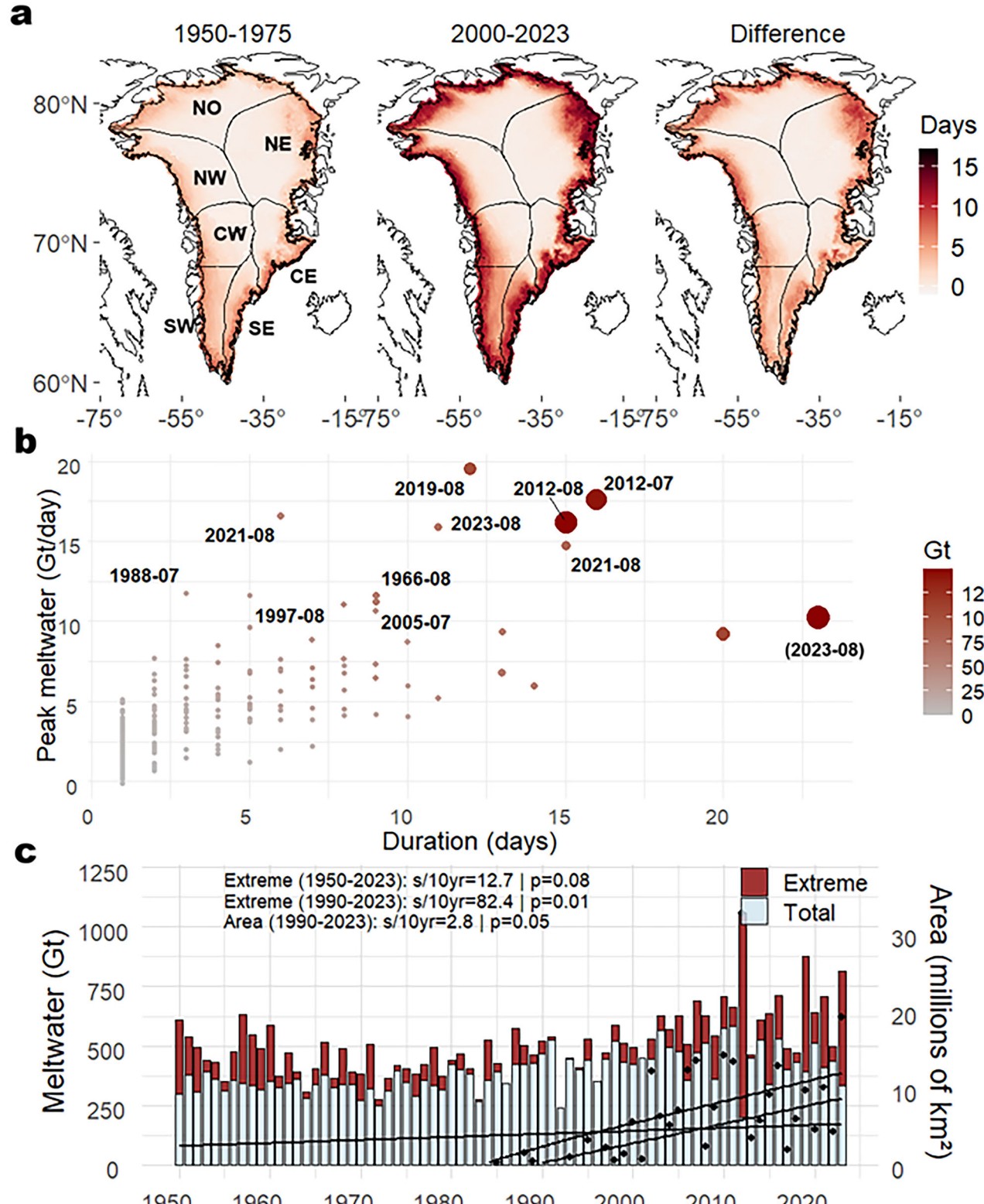

**Fig. 1 | Changes in the magnitude, spatial extent, and frequency of extreme surface melting events over the Greenland ice-sheet (GrIS).** Average number of extreme melting days during summer for the 1950–1975 and 2000–2023 periods, along with their difference (**a**). Magnitude–duration relationship of extreme melting events, with detrended peak daily meltwater (-D) on the *y*-axis and event duration on the *x*-axis; point size reflects total accumulated meltwater (-D) (**b**). Note that the late July–early August 2023 event is excluded from the main analysis, as it falls outside the top 10 extreme melting events ranked by total meltwater production. However, it is referenced in parentheses because it sets a record

(1950–2023) for the longest duration of extreme meltwater production. Temporal trends in summer meltwater accumulation, separated into extreme (red) and non-extreme (Total, light blue) days (**c**). The s/10 yr indicates the slope per decade. The solid black lines represent the regression lines for total summer meltwater accumulation (1950–2023 and 1990–2023) and total area, calculated by the cumulative sum of daily meltwater pixels affected by extreme melting events each season (1985–2023). The left *y*-axis shows simulated meltwater (bars), and the right *y*-axis indicates the total area (black points), derived from passive microwave satellite observations.

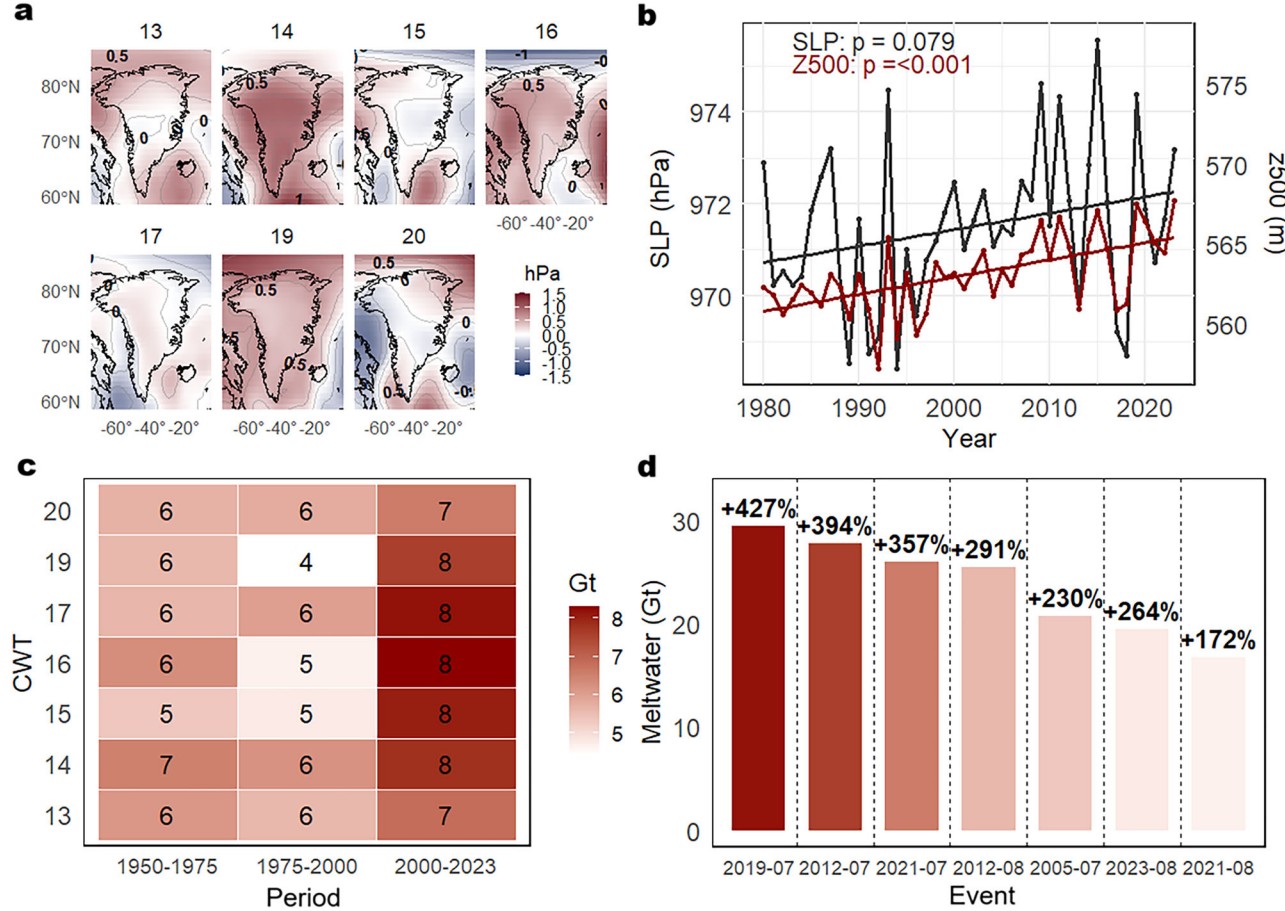

**Fig. 2 | Atmospheric circulation changes and associated meltwater anomalies during Greenland ice-sheet (GrIS) extreme melting events.** Average sea level pressure (SLP) patterns for the Circulation Weather Types (CWTs) related to the recent extreme melting events (2000–2023) (**a**). Positive values (red) and negative (blue) anomalies relative to the 1950–1975 period. Trends in SLP and geopotential height at 500hPa (Z500) over Greenland from 1980 to 2023 (**b**). The solid black lines represent the regression lines. Average daily meltwater associated with each CWT across three time periods, illustrating an increase in melt linked to the same CWT over time (**c**). Meltwater anomalies during recent (2000–2023) extreme melting events, expressed as a percentage relative to the mean meltwater for each CWT during the 1950–1975 reference period (**d**).

ensemble size = 10). Although the MAR-CMIP6 ensemble displays consistent trends, model uncertainty remains high, with projected increases ranging from +200% to +500%, as indicated by the error bars in Fig. 5c. A detailed analysis of individual model averages from MAR-CMIP5 and MAR-CMIP6 is provided in Figs. S6 and S7. The multi-model means across both CMIP phases indicate that the most significant increases are projected for northern Greenland, particularly in the NE (+690%) and NO (+563%) sectors—regions recently affected by the most extreme melting events (Fig. 5).

## Discussion

Satellite observations and regional climate modeling reveal a consistent and significant increase in extreme melting events over the GrIS, highlighting three principal dimensions of recent change. (1) There has been a substantial expansion in the spatial footprint of extreme melting, with a growing impact on high-elevation areas previously unaffected by surface melting—equating to an increase of approximately 2.8 million km² per decade according to satellite observations (Fig. 1a). (2) The majority of the top 10 most extreme melting events—defined by peak daily meltwater production, event duration, and total meltwater production—have occurred in the recent 2000–2023 period (7 out of 10). These events also show a clear temporal shift and an increase in frequency, with extremes increasingly extending into late summer. For instance, the August 2023 event—excluded from the main analysis because the ranking was based on the

peak meltwater production per event—lasted 23 consecutive days, setting a new record for duration (Fig. 1b). (3) The magnitude of extreme melting events has intensified, with meltwater production far exceeding past periods. Since 1990, the trend in extreme summer meltwater volume has increased sixfold—from 12.7 Gt per decade during 1950–2023 to 82.4 Gt per decade in the 1990–2023 period (p = 0.01). Although several extreme melt events occurred during the pre-1975 period, only one exceeded the peak daily meltwater threshold used to define extreme melting events. This indicates that their magnitudes were generally lower than those observed in recent decades. This intensification underscores the rapid acceleration of surface melt since the 1990s under ongoing climate change[3,4,17]. The magnitude of change observed in specific extreme events substantially exceeds what is recorded in paleoclimate records. The July 2012 event exemplifies this, with paleoclimate data from the EastGRIP ice core indicating that a comparable event occurred only once during the Holocene, in 986 CE [38]. Other sources report similar episodes in 1889 and during the Medieval Warm Period, over seven centuries earlier[9].

A precise partitioning of thermodynamic and dynamically induced changes reveals that the top three extreme melting events (August 2012, July 2019, July 2021) have no dynamic precedents within the analog search, highlighting the absence of historical counterparts with similar atmospheric-circulation drivers. The analog analysis of the top 10 extreme melting events shows that favorable synoptic conditions now occur 0.24 days more frequently per decade, based on the

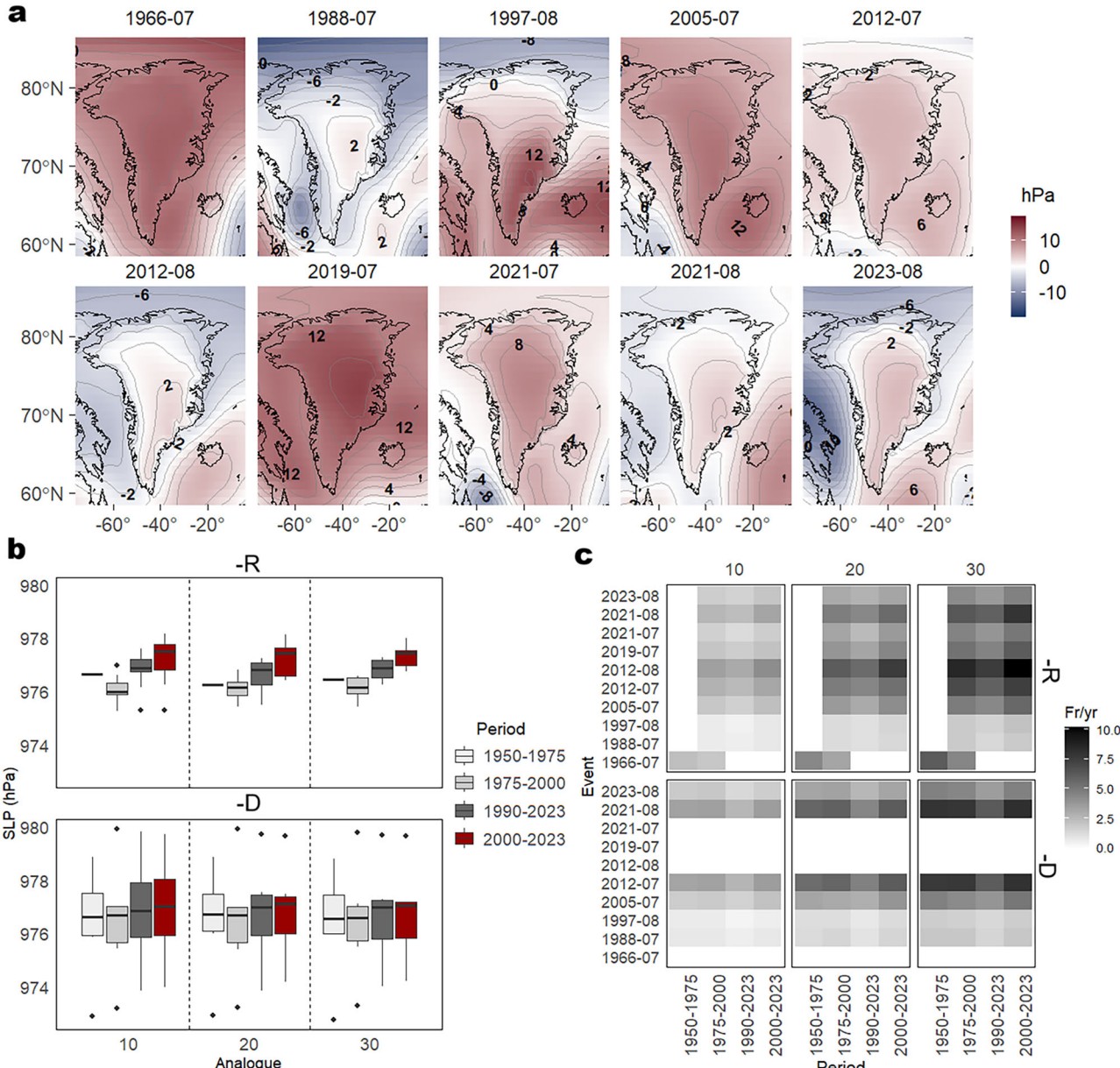

**Fig. 3 | Synoptic configurations of the top 10 extreme melting events over Greenland.** Spatial anomalies of sea level pressure (SLP) for each of the top 10 extreme melting events, relative to the 1950–1975 period, with values indicating the anomaly in hPa for each event (**a**). Average SLP of analogs across four periods (1950–1975, 1975–2000, 1990–2023, and 2000–2023), with boxes representing distributions based on 10, 20, and 30 analogs per event (**b**). Frequency of analogs per year (fr/yr) for each event and period, computed for both raw and detrended SLP datasets (**c**). A complete description of the melt events, including timing, peak daily meltwater production, and total accumulated meltwater, is provided in Table S1.

30 closest analogs identified for the 1990–2023 period, although this trend is not statistically significant (p > 0.05) (Fig. S5). This SLP trend is consistent with observed changes in Z500 and T850 anomalies over the same timeframe (Fig. S3), indicating mid-tropospheric warming and an increased frequency of atmospheric blocking patterns over the GrIS[6,14,16,39,40]. These findings demonstrate that using either SLP or Z500 for synoptic classification yields consistent results due to their strong correlation with mid-tropospheric warming. Arctic amplification is changing blocking patterns in ways that favor more persistent high-pressure conditions over Greenland[41,42], and climate change is altering the fundamental dynamics of atmospheric blocking patterns[13,39]. Particularly, the increase of omega-blocking patterns has been linked to anomalously low spring snow cover conditions over North America[13].

While dynamic atmospheric changes have set the stage for extreme melting events, here we quantify the thermodynamic forcing

that has intensified meltwater production and extreme melting events independently of the dynamic changes. Attribution analysis based on 10 analogs with SLP-D shows that mean daily meltwater production during these events increased by an average of 25% compared to their 1950–1975 analogs. This increases to a 63% intensification of meltwater production when extreme melting events without analogs are included. This thermodynamic signal exhibits significant regional variability, with the most pronounced increases observed in the NW (+75%) and NE (+78%) sectors of the GrIS. The enhanced meltwater production—independent of changes in circulation weather types (CWTs; Figs. S3 and S4)—is linked to multiple surface energy balance processes that have collectively lowered the threshold for extreme melting across much of the GrIS. These include the expansion of melt-prone areas and an increase in available energy for melting, driven by increases in both longwave, shortwave radiation, as well as sensible

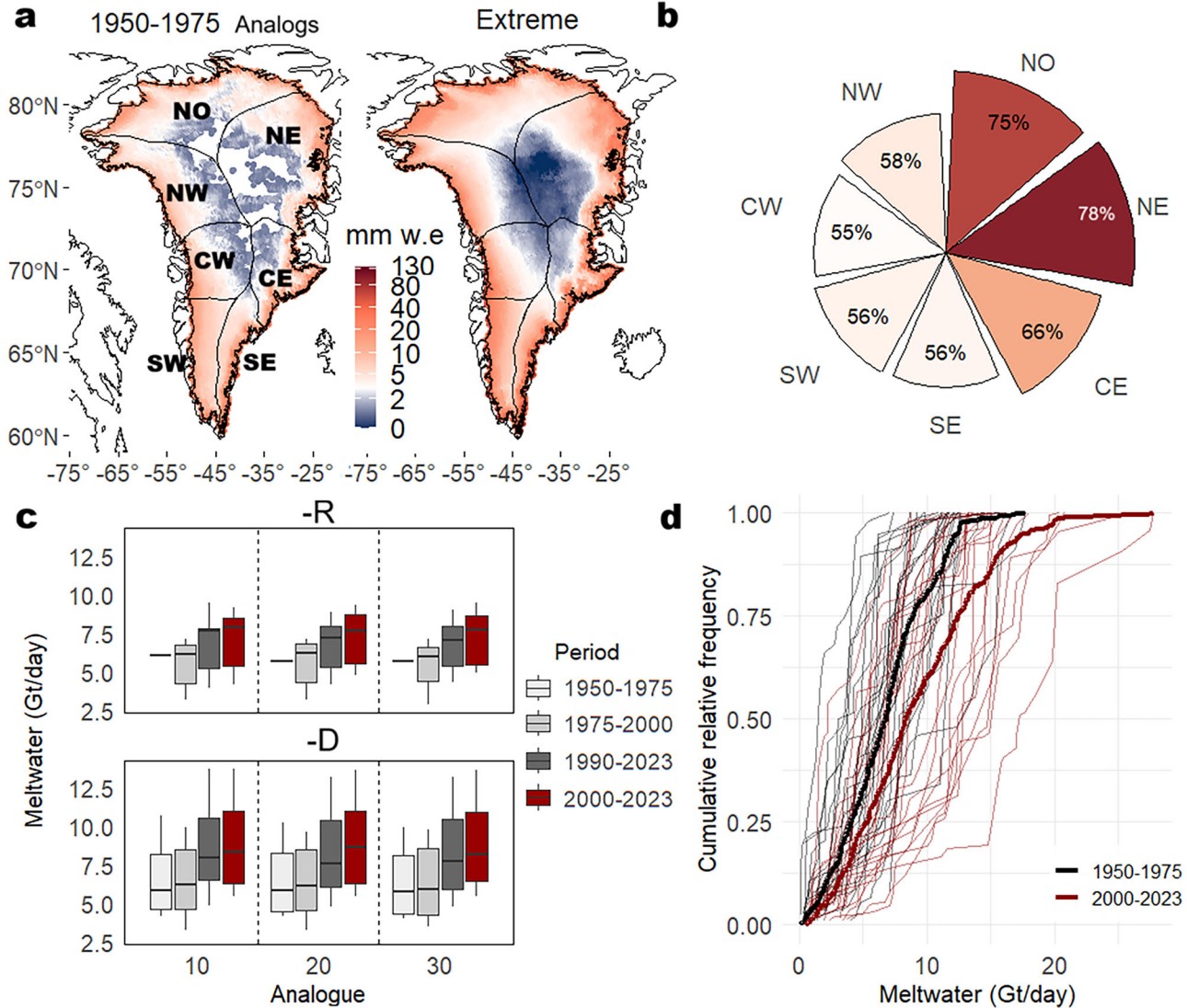

**Fig. 4 | Thermodynamic increase of meltwater during extreme melting events.** Average daily meltwater over the GrIS during the 10 analogs for the 1950–1975 period sea-level pressure detrended (SLP-D and 10 analogs) and during the top 10 extreme melting events (**a**). Anomaly between the average daily meltwater of the 10 analogs -D (1950–1975) and all the top 10 extreme melting events (**b**). Comparison of average daily meltwater over the GrIS from the 10 analogs using sea-level pressure raw (SLP-R) and SLP-D data across different periods (**c**). Cumulative relative frequency of the 10 analogs -D from 1950–1975 and 2000–2023 (**d**). A full description of the extreme melting dates, peak daily meltwater, and total accumulated meltwater for each event is provided in Table S1.

heat fluxes[6]. In particular, surface albedo decline resulting from snowline retreat and the accumulation of light-absorbing impurities has been especially important in the south-western (SW) region, where increased net shortwave radiation has triggered a positive feedback loop that amplifies melting[20]. During omega-blocking situations, increased cloud cover and the frequent transport of warm, moist air masses lead to amplified downward longwave radiation and surface melt in the northern and western GrIS regions, whereas sunnier conditions prevail in southern regions[17]. The two largest record-breaking peak meltwater events—July 2012 and July 2019—occurred under persistent blocking regimes characterized by strong positive SLP anomalies and sustained surface melting[43]. In contrast, the third largest peak meltwater event, in August 2021, illustrates a different pathway to extreme melting driven by atmospheric river–induced moisture transport[44]. This event caused intense surface melt conditions over western Greenland and even resulted in rainfall at the GrIS summit[45]. During this synoptic configuration, a high-pressure system over the eastern flank of Greenland transports warm, moist clouds toward western Greenland, enhancing sensible heat fluxes and downwelling longwave radiation. Meanwhile, reduced cloud cover in the eastern sectors allowed increased incoming shortwave radiation, further amplifying the melt potential across the ice sheet[17].

Future projections indicate a three-order-of-magnitude increase in July and August extreme melting, with anomalies of +234 % (MAR-CMIP5) and +372 % (MAR-CMIP6) based on 10 General Circulation Models (GCMs; n = 10) with respect to the IPCC AR6 polar regions baseline climate towards 2090-2100. This transition to a regime of recurrent, high-intensity melt events has relevant implications for GrIS mass balance and global sea-level rise. Emerging processes such as increasing rainfall intensity (i.e., +7.5 Gt month$^{-1}$ and 20.8 mm h$^{-1}$ year$^{-1}$ in GrIS, September 1980-2019)[46] and expanding supraglacial lake area and snow impurities [47] further compound these risks. Ground-based observations during September 2022 show that multiple melt events occurred across the GrIS, with the largest event in central-western Greenland increasing ice velocities by up to 240%; however, these speed-ups were short-lived, and the corresponding annual increase in ice discharge was only ~2%[48]. These observations indicate that the GrIS response to extreme atmospheric forcing can be highly variable, with

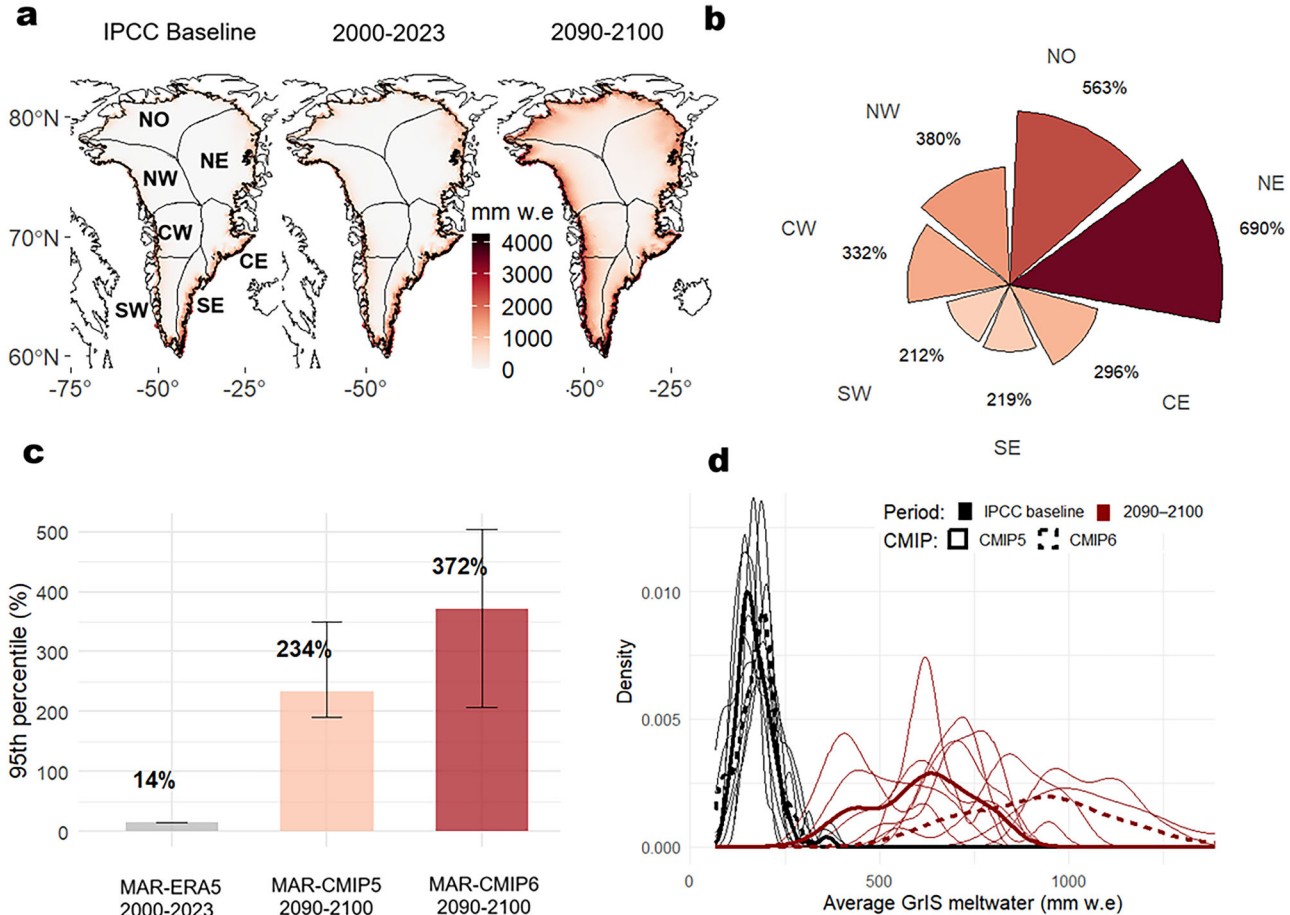

**Fig. 5 | Future intensification of extreme melting events over the Greenland ice sheet (GrIS) by the end of the 21st century.** Accumulated average monthly meltwater during July and August for the Intergovernmental Panel on Climate Change 6th Assessment Report (IPCC AR6) baseline period (1986–2005) and the future period (2090–2100), based on the multi-model mean from Common Management Information Protocol 5 (CMIP5) and CMIP6 ensembles (**a**). Meltwater anomalies between the future and baseline periods, grouped by GrIS regions (**b**). Meltwater anomalies relative to the IPCC AR6 baseline period (**c**). Bars show the anomalies for the historical period (2000–2023), with horizontal lines indicating the maximum and minimum values across CMIP5 and CMIP6 models for the future period (2090–2100). Probability density function of mean GrIS meltwater during July and August for the IPCC AR6 baseline and future periods, shown separately for CMIP5 and CMIP6 (**d**).

transient but substantial impacts on ice motion, which may have implications for local ice-sheet dynamics over short timescales. Despite providing evidence for projected increases in summer extreme meltwater, to our knowledge, no previous study has simulated the frequency and magnitude of GrIS extreme melting events like those observed in July 2012 and July 2019 in future climate scenarios, highlighting a critical gap in our understanding of GrIS summer melting patterns. The meltwater production analyzed in this study could be partially absorbed by the snowpack and firn, thereby reducing runoff and its contribution to sea-level rise[49]. A key process to investigate is the refreezing capacity during these intense melting events. Future research must also address several key uncertainties related to extreme melting events on the GrIS, including—but not limited to—the ice sheet's response to cascading feedbacks such as hydrofracturing, changes in basal hydrology, firn permeability, and supraglacial lake drainage, since these processes can amplify surface melting through enhanced connectivity between the surface and the ice-sheet base[20-24,50]. Late-season extremes, such as the August 2023 event, are particularly intensified because they occur when surface albedo is at its seasonal minimum, maximizing the GrIS's vulnerability to the melt-albedo feedback. The interaction between this feedback and extreme melting events—alongside emerging atmospheric processes such as rainfall intensification, rain-on-snow, and ice-layer formation—must be further investigated to better constrain future

projections. The evolving statistics of blocking patterns under continued warming, with improved representation of anticyclonic types in Earth system models, will be essential for reducing uncertainties in future GrIS meltwater projections[41,42]. CMIP5/6 projections likely underestimate the risk of future extreme melting due to persistent model biases in representing and simulating Greenland blocking patterns and elevation-dependent feedbacks[41,42], potentially leading to underestimation of the frequency and intensity of future extreme melting events.

This study builds on a state-of-the-art flow-analog method for analyzing the drivers of climate extreme events[28,29]. Within this framework, previous work has treated periods before 1980 as a counterfactual world with relatively weak anthropogenic influence, while the post-1980 period reflects a climate more strongly affected by anthropogenic forcing[37]. The flow-analog method used here separates thermodynamic and circulation-induced changes and cannot formally isolate anthropogenic forcing from the overall climate signal. Addressing this limitation will require model simulations that explicitly separate natural and anthropogenic contributions[28] and should form the basis of future work. Our results are nevertheless consistent with previous studies indicating that post-2000 exceptional temperature extremes over the GrIS arise from the superposition of a long-term warming trend attributed to anthropogenic influence[4].

Within the United Nations Framework Convention on Climate Change (UNFCCC) Loss and Damage framework, the recent intensification of extreme melting provides critical evidence to inform policy discussions on climate change impacts within the cryosphere[29,51–53]. As Arctic temperatures continue to rise at approximately twice the global mean rate in land-areas[54], the thermodynamically-induced surface melt quantified here will likely intensify, potentially pushing the ice sheet toward thresholds associated with nonlinear mass loss[55]. Beyond Greenland, large-scale oceanographic disruptions driven by GrIS freshwater inputs—most notably their impact on the AMOC—pose significant risks to the global climate system[25,26] with far-reaching social and economic implications. The increasing extreme melting events in GrIS underscore the need for enhanced scientific investigation and targeted policy measures to mitigate the impacts.

## Methods

### Regional atmospheric model
The state-of-the-art atmospheric model Modèle Atmosphérique Régional version 3.14 (MAR v3.14) was used with a spatial resolution of 10 km at a daily resolution[56]. MAR v3.14 was forced with 6-hourly ERA 5 reanalysis data[57] to simulate meltwater production. MARv3.14 simulates melt by solving the prognostic meteorological equations and a cloud microphysical model, coupled with the Soil Ice Snow Vegetation Atmosphere Transfer (SISVAT) model[56,58]. MAR has been extensively validated over the GrIS against satellite-derived melts and meteorological observations[31,42,56,59]. The variable analyzed here is meltwater production, simulated using the SISVAT module of MAR v3.14 coupled with the CROCUS model[60]. CROCUS is a physics-based, one-dimensional, multi-layer snow and ice model that simulates the energy and mass balance of the snowpack, including detailed snow processes such as refreezing, densification, and metamorphism. Albedo is interactively computed in CROCUS based on snow grain properties and these snowpack processes, allowing it to evolve consistently in both historical and future simulations. Meltwater production provides a spatially resolved measure of surface melting that can be directly linked to atmospheric circulation. This approach minimizes uncertainties related to firn processes and allows robust comparison with satellite-derived observations, supporting the attribution of extreme melt events.

### Passive microwave remote sensing
To evaluate the robustness and consistency of simulated daily MARv3.14 meltwater production data for extreme melting events, we conducted a parallel analysis of extreme melting events using daily satellite passive microwave observations. Daily GrIS melt extent from 1979 to 2023 was derived for the melt season from passive microwave records during June to August. Data from the Scanning Multichannel Microwave Radiometer (SMMR) were available every other day from April 1979 to July 1986 due to the sensor's limited swath coverage. Since July 1986, daily melt observations have been provided by the Special Sensor Microwave/Imager (SSM/I) and the Special Sensor Microwave Imager/Sounder (SSMIS), offering improved spatial and temporal coverage. All data were gridded to a 25 km equal-area projection. Melt detection was performed using the cross-polarized gradient ratio (XPGR) method, which identifies the presence of liquid water when a threshold is exceeded in the 19 and 37 GHz brightness temperatures (Tb) at horizontal (H) and vertical (V) polarizations[61]. The data are aggregated at the daily scale over the GrIS.

### Extreme melting classification
GrIS melting episodes were analyzed for the summer season (June, July and August) over the period 1950–2023. While late-season extreme melt events outside these months may be important, our focus on the core summer months captures the majority of extreme melt and ensures comparability of synoptic conditions. A melting day was defined as one in which meltwater production reached or exceeded 1 mmWE day$^{-1}$ at grid-scale[6,62,63]. To detect anomalously strong melt events independent of long-term increases in mean meltwater, simulated daily meltwater production (MAR v3.14) and daily GrIS melt area (passive microwave remote sensing observations) were detrended. Detrending was performed by subtracting the linear regression of total GrIS meltwater and melt area extent, respectively, for each day, thereby removing the influence of rising mean values. The selection of the top 10 extreme melting events was based on the highest total accumulated detrended meltwater across the entire ice sheet. Extreme melting days were defined as those exceeding the 95th percentile of melt values calculated over a 20-year baseline climate period (1986–2005, included). This period was selected according to the IPCC AR6 reference period for polar regions[64] and consistent with GrIS extreme meltwater and surface mass balance (SMB) analysis [17,65]. Days were classified as part of the same extreme melting episode when there was an uninterrupted sequence of at least 1 day with values exceeding the 95th percentile threshold. Each extreme melting episode was characterized by its start and end dates, duration (consecutive days), and assigned a unique identifier. This approach enabled the isolation and detailed analysis of extreme meltwater events independent of the underlying meltwater trends. Linear regression was used to assess trends in meltwater volume and spatial extent, with statistical significance assessed using $t$-tests. The regional analysis of GrIS is performed using a regionalization based on ice-flow characteristics[66].

### Flow analogs
Attribution of extreme melting events is performed using a flow-analog methodology[33,67,68]. This method isolates dynamic (circulation-driven) and thermodynamic (warming-driven) contributions to extreme melting events by holding synoptic conditions constant while allowing background climate to vary between periods. Synoptically similar events are identified by calculating the root mean square difference (RMSD) of daily sea level pressure (SLP) patterns over the Greenland domain (50°N–90°N, 120°W–10°E) during July and August (1950-2023) from ERA5, through an event time window of 20 days around the event. The analog search is performed using 100 bootstrap iterations to generate a set of synthetic similar events after the selection of the number of analogs. Each stochastic simulation consists of randomly picking one analog for each day. SLP is compared with geopotential height at 500 hPa (Z500, m) to assess the sensitivity to variable selection. Three sensitivity analyses assess the robustness of results: (1) For thermodynamics isolation, SLP is detrended at the grid-cell scale by subtracting the daily linear trend (1950-2023) from the original SLP data. This process ensures that the daily differences in meltwater are calculated between analogous synoptic events, independently of the increase in anticyclonic conditions and SLP. (2) Atmospheric circulation analogs for each top 10 extreme melting event are selected based on RMSD similarity, testing 10, 20, and 30 analogs per event; (3) Temporal variability is examined across three approximately 25-year periods (1950–1975, 1975–2000, 2000–2023) and 1990–2023 to account for the increase in meltwater since the 1990s, with 1950–1975 serving as a minimal-warming baseline, while subsequent periods reflect increasing anthropogenic forcing. A fourth temporal period (1990–2023) is additionally included to capture the accelerated GrIS meltwater increase since the 1990s. The near-30-year periods are consistent with the normal climate periods recommended by the World Meteorological Organization to capture natural climate variability. Daily SLP and GrIS meltwater from reference events are compared against their analogs across these periods using both detrended (− D) and non-detrended (− R) SLP data. The differences in meltwater between periods quantify the thermodynamic contribution. The statistical significance of trends is assessed via $t$-tests (p-value). This comprehensive approach provides robust uncertainty estimation for both the flow-analog method and parameter sensitivity.

## Synoptic classification

The increase in meltwater under consistent synoptic conditions was analyzed using a synoptic classification method. This approach allowed us to quantify the thermodynamic contribution to meltwater under the same synoptic conditions, as well as for extreme events without suitable analogs. Specifically, the Extreme Score synoptic classification[69], integrated into the COST733 European project based on synoptic circulation methods[70] was implemented using the SynoptReg R package[71]. The classification focused on the GrIS region (50°N–90°N, 80°W–10°E) during the July–August period from 1950 to 2023, using ERA5 SLP data at daily temporal resolution. Note that June is excluded due to the absence of any top 10 extreme melting events.

The method applies Principal Component Analysis (PCA) to SLP data, retaining 10 principal components (PCs) that capture most (85%) of the SLP variability. Following PCA, a K-means clustering technique is applied using an S-mode matrix with varimax rotation on the retained PCs. Each PC is then used to derive two circulation weather types (CWTs), identified by their highest positive and negative correlation values, resulting in 20 CWTs derived from the 10 retained PCs. The days on which the peak daily meltwater of the top 10 extreme melting events was reached were associated with their corresponding CWTs. The average daily accumulated meltwater for these CWTs was calculated across the periods of 1950–1975, 1975–2000, 1990–2023 and 2000–2023. This approach provided a complementary analysis to the flow analogs and enabled the quantification of meltwater increases independently of dynamic factors. SLP and Z500 are compared with 850 hPa air temperature (T850, °C) to assess the relationship between pressure patterns and mid-tropospheric warming, with Mann-Kendall trend analyses and associated p-values used to evaluate significance.

## Projected increase of extreme melt

The historical evolution of extreme meltwater thresholds was contextualized using MAR v3.9 projections for the late 21st century (2090–2100) under a high-end emission scenario (CMIP5 RCP8.5 and CMIP6 SSP5-8.5)[31]. MAR v3.9 meltwater production, SISVAT and CROCUS coupled simulation, including albedo simulations, snow grain properties and snowpack physics. This analysis focused on calculating the monthly extreme meltwater threshold (95th percentile) difference from the IPCC AR6 baseline climate period (1986–2005) and the 2090–2100 period for each model and CMIP phase. Thus, anomalies are subtracted between temporal periods and for the same model configuration, and therefore, the MAR version does not affect the results. The CMIP5 (MAR-CMIP5 hereafter) ensemble included Had-GEM2-ES, MIROC5, NorESM1-M, ACCESS1.3, CSIRO-Mk3-6-0, and IPSL-CM5A-MR, while the CMIP6 (MAR-CMIP6 hereafter) ensemble included CESM2, CNRM-CM6-1, CNRM-ESM2-1, MRI-ESM2-0, and UKESM1-0-LL. These models were dynamically downscaled using MAR v3.9 at a spatial resolution of 15 km and a temporal resolution of 6-h[31]. Detailed information on the downscaling methodology, projection protocols, and model selection criteria is available at Hofer et al.[31].

## Data availability

Data that supports the findings of this work is publicly available. ERA5 reanalysis data are available at https://cds.climate.copernicus.eu/#!/search?text=ERA5. MAR-ERA5 outputs can be downloaded at http://ftp.climato.be/fettweis/. MAR-CMIP5 and MAR-CMIP6 outputs can be downloaded at http://ftp.climato.be/fettweis/MARv3.9/ISMIP6. Satellite passive microwave data is available at https://nsidc.org/ice-sheets-today (last accessed: 10/06/2025).

## Code availability

Codes to reproduce the flow analogs attribution methodology are available at https://github.com/lemuscanovas/climattR. The codes for the synoptic classification can be found at https://github.com/lemuscanovas/synoptReg (last accessed: 10/06/2025). Codes to reproduce the findings are available upon request from the first author.

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

## Acknowledgements

This work falls within the NEOGREEN (PID2020-113798GB-C31), HIGH-ARCTIC (PID2023-146730NB-C31), and GRELARCTIC (PID2023-146730NB-C31) projects from the ANTALP research group (2021-SGR-00269). J.B. thanks the Ministerio de Ciencia, Innovación y Universidades, Spain (PRE2021-097046), and M.O. thanks ICREA (AGAUR–Generalitat de Catalunya) for financial support.

## Author contributions

J.B. wrote the paper and produced the results. J.B., S.G., and M.L.C. conceived the work. X.F. performed the MAR simulations. M.O. secured funding acquisition. J.B., S.G., M.L.C., X.F., J.I.L.M., and M.O. provided feedback and contributed to the interpretation of the results.

## Competing interests

The authors declare no competing interests.
