## [Transparent Peer Review file · Nature Communications]

Record-Breaking Greenland Ice Sheet Melt Events under Recent and Future Climate

Corresponding Author: Dr Josep Bonsoms

Version 0:

Reviewer comments:

Reviewer #1

(Remarks to the Author)

This is my second review of this paper, having read it for a different journal. I would like to thank the authors for responding so thoroughly to my previous comments. While the authors have dealt with the majority of my queries satisfactorily, there is one main and several minor outstanding issues which I outline below.

Main point

1. Concern about attributing all of the change in the thermodynamic drivers to anthropogenic climate change. As I understand it, the circulation-conditioned flow-analogue framework enables the authors to separate out large-scale circulation drivers from thermodynamical drivers. However, there is then an implicit assumption that all or the vast majority of the impact of these drivers is due to anthropogenic climate change. Does this not ignore other non-anthropogenic (i.e. natural) potential impacts on thermodynamic drivers such as the North Atlantic Oscillation (e.g. <https://www.nature.com/articles/s41612-024-00686-2>)? Additional explanation to clarify this for those not well-versed in the climate change attribution literature might be a useful addition.

Minor points

L52: There is an extra space.

L63: You revert back to runoff here without clearly stating the difference compared to meltwater production. Perhaps "An increase in the amount of meltwater produced that leaves the ice sheet (runoff) could ..." would be clearer?

L80: Missing space.

Reviewer #2

(Remarks to the Author)

Summary of the study:

The study "Thermodynamic amplification of extreme Greenland Ice Sheet melt events attributable to anthropogenic climate change" aims to attribute the 10 most extreme Greenland ice sheet melting events from 1950 to 2023 by decomposing the relative contributions of thermodynamic versus dynamic drivers. The idea is to address the question: how much of the intensification of extreme melt is due to direct surface warming v/s changes in atmospheric circulation patterns that favour melt. Furthermore, the authors use CMIP5/CMIP6 climate models to project the future intensification of extreme melt events under high-emission scenarios (RCP8.5, SSP5-8.5) to 2100, with a particular focus on identifying regional hotspots, especially in northwest and northern Greenland, where the risk of extreme melt accelerates most.

I think the study is thorough and well written in general. The two reviewers' have rightly pointed out a few shortcomings and improvements. My comments to the authors' response to the original reviews is appended below. Additionally, I recommend the authors to elaborate a bit more on the methodology in the last paragraph of the introduction (e.g. analogue based approach), as in the current version, the information on methods is not enough for diverse readers to continue reading through to the results section.

Comments on authors' response to Reviewer #1

On comment #1

I believe that the authors have made improvements, like clarifying the title to emphasize thermodynamic amplification and softening language in the abstract. However, it appears that they have perhaps sidestepped the reviewer's core concern rather than addressing it. The reviewer's point was not that flow analogue methods are invalid, but rather the claim "attribution to anthropogenic climate change" without performing the model-based attribution (likelihood in all-forcing vs. natural-only scenarios) that such claims require. The authors' response instead defends their method by citing IPCC AR6, which is not the point here as showing that analogue methods are established science does not prove they constitute formal anthropogenic attribution. I understand that adding a new analysis is tedious, but perhaps the authors may explicitly state (wherever they may need to) that this is conditional attribution and does not include model-based likelihood comparison.

On comment #2

The authors' choice to focus on meltwater production is defensible, I believe. They have now added a sentence clarifying why and referenced that runoff trends follow meltwater trends. I think it is acceptable that the authors' don't add runoff analysis, but they transparently justify the choice and acknowledge the limitation.

Comments on authors' response to Reviewer #2

On comment #1

The reviewer is arguing that the methods and datasets are standard, not novel. The authors' response argues that novelty lies in the integration and application to GrIS extreme melting and not the methods themselves and distinguishes their work from prior studies. I think that while the response does clarify what's new (first attribution framework for GrIS extremes, etc.), calling this a "frontier identified by IPCC AR6" may overstate the impact as the core methods remain standard.

On comment #2

The study assumes that all changes after 1975 are anthropogenic, ignoring that natural multi-decadal oscillations (Atlantic Multidecadal Oscillation, etc.) contribute to observed warming. The reviewer has pointed this out and cites a fire-weather study where one-third of the signal was natural variability - showing why this is important. The authors' response is again citing IPCC AR6 in recognition of flow analogue methods, but none of these steps actually separate natural from forced variability. While the authors acknowledge that natural variability "cannot be fully excluded" and intend to emphasize these uncertainties in the revised discussion, they may need to explicitly limit claims to thermodynamic response to observed warming (without claiming anthropogenic causation).

Reviewer #3

(Remarks to the Author)

Thermodynamic amplification of extreme Greenland Ice Sheet melt events attributable to anthropogenic climate change.

This paper is analyzing the evolution of extreme surface meltwater events over the Greenland ice sheet since 1950 and into the future. Extreme meltwater events in the last decade have no analogues that can be found in earlier times, and most of the most extreme of these events took place in this past decade and are associated with a change in atmospheric circulation due to anthropogenic changes on our climate.

This is a second round of reviews, and I appreciate the effort made by the authors to address the comments of both reviewers.

The paper is well written with convincing supporting evidence. I would welcome its publication after minor revisions are addressed.

General comments

1. It would be helpful to the reader to refer to the method section for each concept you have described there (extreme melting classification, flow analogues, ...).
2. When you talk about the top 10 extreme melting events, I am very surprised that year 1997 with about 25 Gt of total meltwater (based on figure 1) made the list as opposed to say year 1957 with a total of over 250 Gt of meltwater. (I also think there is a typo in table S1 about that year.)
3. In figure 1, it looks like the time period 1950-1962 experiences a series of strong melting events that are in magnitude

similar to the ones spanning the period 2000-2023 (except for the years 2012, 2019, and 2023). Please add a few lines in your discussion about this time and discuss how the melting event then differed from the ones in the last decades (none of these years made the cuts of the top 10 events despite large extreme events).

Line by line comments (The referencing is based on the updated manuscript)

Line 42: replace "extreme melting events" with (for example) "extreme melting events (see Methods section)".

Line 68: "No study" – It is a matter of style, but I find this semantic unnecessarily provocative and subjective, and it does not age well! Please rephrase with something less categorical such as: "To our knowledge, no study ...", "Few if any study has...".

Line 85: replace "summer" with "summer (June, July, and August)"

Line 113: "it set" -> "it sets".

Line 119: replace "A synoptic classification" with "A synoptic classification (see Methods section)" (or something similar).

Line 157: replace "a flow analogue method" with "a flow analogue method (see Methods section)" (or something similar).

Line 159: please define "-R".

Line 160: please explain how you determined these top 10 extreme melting events.

Line 282: what does "n=10" refer to?

Line 294: similar remark as for line 68, "no existing study", that you know of! You could also say something like "these events are understudied" or something similar.

Line 315: define UNFCCC.

Figures and tables

Figure 1: Panel c:

- What does the 's' mean in s/10yr? Please explain in the caption.
- What are the black dots on the figure added since 1985? Please add to the caption.
- What are the black lines on the figure? It looks like they are the decadal trends. If so, say so in the caption.
- Since you are comparing the time spans 1950-1975 and 2000-2023 in panel a, it would be informative to have these slopes per 10yr as well.

Figure 2: Panel b: why start the time series in 1980 when your analysis starts with year 1950?

Figure S2: The subpanel associated with CWT 13 shows a decreasing red line meaning the slope/10yr should be negative and read -20.959.

Table S1: the total meltwater for year 1997 looks more like 25 Gt (based on Figure 1).

Version 1:

Reviewer comments:

Reviewer #1

(Remarks to the Author)

I thank the authors for thoroughly addressing my previous review. I have just a couple of remaining minor points.

L3: There should be a space between the end of the title and the lead author.

L16: I think 'of' would be better than 'in'.

L50: if these factors enhanced the rise, it might be worth stating briefly what caused the underlying rise.

L224: It seems to be Figure 1c that shows the temporal trend. Also, I think it would be worth stating clearly (in the figure caption, maybe) that in this case 'area' is the sum of the area experiencing melting each day per year (I think). Otherwise it could be misleading that the decadal trend (2.8M km²) in melt area is far greater than the total area of the ice sheet (~1.8M km²).

Reviewer #2

(Remarks to the Author)

I am happy with the revisions.

Reviewer #3

(Remarks to the Author)

I would like to thank the authors for their careful considerations in replying to all the reviewers comments. I personally have no other comments on this study. Once the comments from the other reviewers are addressed, I believe this work would be ready for publication.

Reviewers Comments:

Reviewer #1 (Remarks to the Author):

This is my second review of this paper, having read it for a different journal. I would like to thank the authors for responding so thoroughly to my previous comments. While the authors have dealt with the majority of my queries satisfactorily, there is one main and several minor outstanding issues which I outline below.

We would like to sincerely thank the reviewer for his/her evaluation of our work.

Please find below a point-by-point response to the comments, along with the corresponding changes made in the manuscript.

Main point

1. Concern about attributing all of the change in the thermodynamic drivers to anthropogenic climate change. As I understand it, the circulation-conditioned flow-analogue framework enables the authors to separate out large-scale circulation drivers from thermodynamical drivers. However, there is then an implicit assumption that all or the vast majority of the impact of these drivers is due to anthropogenic climate change. Does this not ignore other non-anthropogenic (i.e. natural) potential impacts on thermodynamic drivers such as the North Atlantic Oscillation (e.g. <https://www.nature.com/articles/s41612-024-00686-2>)? Additional explanation to clarify this for those not well-versed in the climate change attribution literature might be a useful addition.

We are grateful for this comment, and we have carefully addressed this point and added further explanations throughout the manuscript.

We clarified the objective (separating thermodynamic and circulation effects using flow analogues) and removed statements attributing changes to anthropogenic climate change, as suggested by Reviewer 1.

We added in the Discussion section:

“... This study builds on state-of-the-art flow-analogue method for analyzing the drivers of climate extreme events^{28,29}. Within this framework, previous work has treated periods before 1980 as a counterfactual world with relatively weak anthropogenic influence, while the post-1980 period reflects a climate more strongly affected by anthropogenic forcing³⁷. The flow-analogue method used here separates thermodynamic and circulation-induced changes, and cannot formally isolate anthropogenic forcing from the overall climate signal. Addressing this limitation will require model simulations that explicitly separate natural and anthropogenic contributions²⁸ and should form the basis of future work. Our results are nevertheless consistent with previous studies indicating that post-2000 exceptional temperature extremes over the GrIS arise from the superposition of a long-term warming trend attributed to anthropogenic influence⁴...”

⁴: Hörhold, M., Münch, T., Weißbach, S. *et al.* Modern temperatures in central–north Greenland warmest in past millennium. *Nature* 613, 503–507 (2023). <https://doi.org/10.1038/s41586-022-05517-z>

Regarding statements on anthropogenic climate change, we have modified them accordingly:

Title:

“Thermodynamic amplification of extreme Greenland Ice Sheet melt events attributable to anthropogenic climate change”

To:

“Record-Breaking Greenland Ice Sheet Melt Events under Recent and Future Climate”

Abstract

From:

“... Despite its central role in global sea-level rise and Earth system dynamics, attribution of these extreme melting events to climate change has remained absent.”

To:

“...Despite its central role in global sea-level rise and Earth system dynamics, the mechanisms driving extreme melting events in the GrIS remain incompletely understood. Here, we analyze extreme melting events over 1950–2023, disentangling thermodynamic and dynamic contributions using a novel analogue-based framework coupled with a regional climate model”.

From:

“...By isolating the thermodynamic contribution, we determine how much of the intensity of the 10 most extreme melting events can be attributed to thermodynamic processes linked to anthropogenic climate change.”

To:

“...By isolating the thermodynamic contribution, we quantify how thermodynamic processes have intensified the 10 most extreme melting events, independent of changes in circulation”.

Introduction

“...To our knowledge, no study has yet quantified how anthropogenic climate change has amplified meltwater production during extreme events in GrIS...”

To:

“... To our knowledge, no study has yet quantified the partitioning of thermodynamic and circulation contributions to meltwater production during extreme events in the GrIS...”

Minor points

L52: There is an extra space.

Corrected.

L63: You revert back to runoff here without clearly stating the difference compared to meltwater production. Perhaps "An increase in the amount of meltwater produced that leaves the ice sheet (runoff) could ..." would be clearer?

Added.

L80: Missing space.

Corrected.

Reviewer #2 (Remarks to the Author):

Summary of the study:

The study “Thermodynamic amplification of extreme Greenland Ice Sheet melt events attributable to anthropogenic climate change” aims to attribute the 10 most extreme Greenland ice sheet melting events from 1950 to 2023 by decomposing the relative contributions of thermodynamic versus dynamic drivers. The idea is to address the question: how much of the intensification of extreme melt is due to direct surface warming v/s changes in atmospheric circulation patterns that favour melt. Furthermore, the authors use CMIP5/CMIP6 climate models to project the future intensification of extreme melt events under high-emission scenarios (RCP8.5, SSP5-8.5) to

2100, with a particular focus on identifying regional hotspots, especially in northwest and northern Greenland, where the risk of extreme melt accelerates most.

I think the study is thorough and well written in general. The two reviewers' have rightly pointed out a few shortcomings and improvements. My comments to the authors' response to the original reviews is appended below. Additionally, I recommend the authors to elaborate a bit more on the methodology in the last paragraph of the introduction (e.g. analogue based approach), as in the current version, the information on methods is not enough for diverse readers to continue reading through to the results section.

We would like to sincerely thank the reviewer for their careful evaluation of our work.

Please find below a point-by-point response to the comments, along with the corresponding changes made in the manuscript.

Comments on authors' response to Reviewer #1

On comment #1

I believe that the authors have made improvements, like clarifying the title to emphasize thermodynamic amplification and softening language in the abstract. However, it appears that they have perhaps sidestepped the reviewer's core concern rather than addressing it. The reviewer's point was not that flow analogue methods are invalid, but rather the claim "attribution to anthropogenic climate change" without performing the model-based attribution (likelihood in all-forcing vs. natural-only scenarios) that such claims require. The authors' response instead defends their method by citing IPCC AR6, which is not the point here as showing that analogue methods are established science does not prove they constitute formal anthropogenic attribution. I understand that adding a new analysis is tedious, but perhaps the authors may explicitly state (wherever they may need to) that this is conditional attribution and does not include model-based likelihood comparison.

We have carefully addressed this point and added additional explanations throughout the manuscript.

Regarding the methodology, we included further clarification in the Discussion section:

"... This study builds on state-of-the-art flow-analogue method for analyzing the drivers of climate extreme events^{28,29}. Within this framework, previous work has treated periods before 1980 as a counterfactual world with relatively weak anthropogenic influence, while the post-1980 period reflects a climate more strongly affected by anthropogenic forcing³⁷. The flow-analogue method used here separates thermodynamic and circulation-induced changes and cannot formally isolate anthropogenic forcing from the overall climate signal. Addressing this limitation will require model simulations that explicitly separate natural and anthropogenic contributions²⁸ and should form the basis of future work. Our results are nevertheless consistent with previous studies indicating that post-2000 exceptional temperature extremes over the GrIS arise from the superposition of a long-term warming trend attributed to anthropogenic influence⁴..."

⁴: Hörhold, M., Münch, T., Weißbach, S. *et al.* Modern temperatures in central–north Greenland warmest in past millennium. *Nature* 613, 503–507 (2023). <https://doi.org/10.1038/s41586-022-05517-z>

Regarding statements on anthropogenic climate change, we have modified them accordingly:

Title:

"Thermodynamic amplification of extreme Greenland Ice Sheet melt events attributable to anthropogenic climate change"

To:

“Record-Breaking Greenland Ice Sheet Melt Events under Recent and Future Climate”

Abstract

From:

“... Despite its central role in global sea-level rise and Earth system dynamics, attribution of these extreme melting events to climate change has remained absent.”

To:

“...Despite its central role in global sea-level rise and Earth system dynamics, the mechanisms driving extreme melting events in the GrIS remain incompletely understood. Here, we analyze extreme melting events over 1950–2023, disentangling thermodynamic and dynamic contributions using a novel analogue-based framework coupled with a regional climate model”.

From:

“...By isolating the thermodynamic contribution, we determine how much of the intensity of the 10 most extreme melting events can be attributed to thermodynamic processes linked to anthropogenic climate change.”

To:

“...By isolating the thermodynamic contribution, we quantify how thermodynamic processes have intensified the 10 most extreme melting events, independent of changes in circulation”.

Introduction

“...To our knowledge, no study has yet quantified how anthropogenic climate change has amplified meltwater production during extreme events in GrIS...”

To:

“... To our knowledge, no study has yet quantified the partitioning of thermodynamic and circulation contributions to meltwater production during extreme events in the GrIS...”

On comment #2

The authors' choice to focus on meltwater production is defensible, I believe. They have now added a sentence clarifying why and referenced that runoff trends follow meltwater trends. I think it is acceptable that the authors' don't add runoff analysis, but they transparently justify the choice and acknowledge the limitation.

Comments on authors' response to Reviewer #2

On comment #1

The reviewer is arguing that the methods and datasets are standard, not novel. The authors' response argues that novelty lies in the integration and application to GrIS extreme melting and not the methods themselves and distinguishes their work from prior studies. I think that while the response does clarify what's new (first attribution framework for GrIS extremes, etc.), calling this a "frontier identified by IPCC AR6" may overstate the impact as the core methods remain standard.

Changed from:

“...Separating these contributions is therefore crucial to accurately attribute recent increases in extreme climate events to human-induced climate change, addressing a key knowledge gap highlighted by the IPCC AR6³⁰.”

To

“...Separating these contributions is therefore crucial to accurately understand recent increases in extreme climate events within a climate change context, addressing a knowledge gap noted by the IPCC AR6

On comment #2

The study assumes that all changes after 1975 are anthropogenic, ignoring that natural multi-decadal oscillations (Atlantic Multidecadal Oscillation, etc.) contribute to observed warming. The reviewer has pointed this out and cites a fire-weather study where one-third of the signal was natural variability - showing why this is important. The authors' response is again citing IPCC AR6 in recognition of flow analogue methods, but none of these steps actually separate natural from forced variability. While the authors acknowledge that natural variability "cannot be fully excluded" and intend to emphasize these uncertainties in the revised discussion, they may need to explicitly limit claims to thermodynamic response to observed warming (without claiming anthropogenic causation).

We are grateful for this comment, and we have carefully addressed this point and added further explanations throughout the manuscript.

We clarified the objective (separating thermodynamic and circulation effects using flow analogues), introduction and removed statements attributing changes to anthropogenic climate change, as suggested by Reviewer 2.

We clearly indicate the motivation for quantifying the thermodynamic and circulation analysis:

“... Understanding extreme melt events over the GrIS requires disentangling the roles dynamical and thermodynamical factors. While anthropogenic climate change can affect both components, its influence on dynamical drivers is harder to detect due to the large natural variability in atmospheric flows^{28,29}. Separating these contributions is therefore crucial to accurately understand recent increases in extreme climate events within a climate change context, addressing a knowledge gap noted by the IPCC AR6²⁹. Furthermore, projections of future extreme meltwater remain limited, leaving a critical aspect of GrIS mass loss poorly constrained and with still high uncertainty³⁰. To our knowledge, no study has yet quantified the partitioning of thermodynamic and circulation contributions to meltwater production during extreme events in the GrIS, nor assessed the future intensification of such extreme melt.”

“... We disentangle the thermodynamic and dynamic contributions of 10 most extreme GrIS melting events using a novel analogue-based framework^{29,30, 33-37}. Furthermore, we assess how summer (June, July and August) extreme melting magnitudes are projected to evolve under high-emission scenarios (RCP8.5 and SSP5-8.5) from CMIP5 and CMIP6, identifying regional hotspots of intensification toward the end of the 21st century....”

We also modified the Discussion section:

“... This study builds on state-of-the-art flow-analogue method for analyzing the drivers of climate extreme events^{28,29}. Within this framework, previous work has treated periods before 1980 as a counterfactual world with relatively weak anthropogenic influence, while the post-1980 period reflects a climate more strongly affected by anthropogenic forcing³⁷. The flow-analogue method used here separates thermodynamic and circulation-induced changes and cannot formally isolate anthropogenic forcing from the overall climate signal. Addressing this limitation will require model simulations that explicitly separate natural and anthropogenic contributions²⁸ and should form the basis of future work. Our results are nevertheless consistent with previous studies indicating that post-2000 exceptional temperature extremes over the GrIS arise from the superposition of a long-term warming trend attributed to anthropogenic influence⁴...”

Reviewer #3 (Remarks to the Author):

Thermodynamic amplification of extreme Greenland Ice Sheet melt events attributable to anthropogenic climate change.

This paper is analyzing the evolution of extreme surface meltwater events over the Greenland ice sheet since 1950 and into the future. Extreme meltwater events in the last decade have no analogues that can be found in earlier times, and most of the most extreme of these events took place in this past decade and are associated with a change in atmospheric circulation due to anthropogenic changes on our climate.

This is a second round of reviews, and I appreciate the effort made by the authors to address the comments of both reviewers.

The paper is well written with convincing supporting evidence. I would welcome its publication after minor revisions are addressed.

We would like to sincerely thank the reviewer for their careful evaluation of our work.

Please find below a point-by-point response to the comments, along with the corresponding changes made in the manuscript.

General comments

1. It would be helpful to the reader to refer to the method section for each concept you have described there (extreme melting classification, flow analogues, ...).

Corrected.

2. When you talk about the top 10 extreme melting events, I am very surprised that year 1997 with about 25 Gt of total meltwater (based on figure 1) made the list as opposed to say year 1957 with a total of over 250 Gt of meltwater. (I also think there is a typo in table S1 about that year.)

We believe this comment arises from a misunderstanding of the metrics used to define extreme melting events and of the data presented in the figures.

To clarify, Figure 1b shows the daily peak meltwater associated with each event, not the seasonal or total summer meltwater. For the August 1997 event, the daily peak meltwater reaches 16.3 Gt (without detrending) and 11.6 Gt (detrended, as shown in Figure 1b). Table S1 reports these peak daily values as well as the accumulated meltwater during each event, which amounts to 52 Gt (without detrending) and 29.1 Gt (detrended) for 1997. The accumulated values per event are not shown in Figure 1 and are provided only in Table S1 as complementary information on event magnitude.

We confirm that 1957 exhibits a total summer meltwater amount exceeding 250 Gt; however, this is not exceptional, as more than 90% of melt seasons reach similar values (Figure 1c). In 1957, summer melt occurred without any individual daily peak meltwater event ranking among the top 10 extremes. For this reason, 1957 does not appear in Figure 1b or Table S1, which are based exclusively on peak daily meltwater intensity rather than total seasonal melt. Regarding the potential typo in Table S1, we have rechecked the values and confirm that the reported numbers for 1997 are correct.

3. In figure 1, it looks like the time period 1950-1962 experiences a series of strong melting events that are in magnitude similar to the ones spanning the period 2000-2023 (except for the years 2012, 2019, and 2023). Please add a few lines in your discussion about this time and discuss how the melting event then differed from the ones in the last decades (none of these years made the cuts of the top 10 events despite large extreme events).

We have added a brief discussion addressing this point:

L235: "...Although several extreme melt events occurred during the pre-1975 period, only one exceeded the peak daily meltwater threshold used to define extreme melting events. This indicates that their magnitudes were generally lower than those observed in recent decades."

Line by line comments (The referencing is based on the updated manuscript)

Line 42: replace "extreme melting events" with (for example) "extreme melting events (see Methods section)".

Done.

Line 68: "No study" – It is a matter of style, but I find this semantic unnecessarily provocative and subjective, and it does not age well! Please rephrase with something less categorical such as: "To our knowledge, no study ...", "Few if any study has...".

Corrected to "To our knowledge, no study ...".

Line 85: replace "summer" with "summer (June, July, and August)"

Done.

Line 113: "it set" -> "it sets".

Done.

Line 119: replace "A synoptic classification" with "A synoptic classification (see Methods section)" (or something similar).

Done.

Line 157: replace "a flow analogue method" with "a flow analogue method (see Methods section)" (or something similar).

Done.

Line 159: please define "-R".

Done.

Line 160: please explain how you determined these top 10 extreme melting events.

We made reference to the Methods section where a description can be found. We have added: "see Methods section".

Line 282: what does "n=10" refer to?

Changed to: "...based on 10 GCMs (n = 10)".

Line 294: similar remark as for line 68, "no existing study", that you know of! You could also say something like "these events are understudied" or something similar.

Following your previous recommendation, changed to:

"...Despite providing evidence for projected increases in summer extreme meltwater, to our knowledge no previous study has simulated the frequency ..."

Line 315: define UNFCCC.

"... Within the United Nations Framework Convention on Climate Change (UNFCCC) Loss and Damage framework..."

Figures and tables

Figure 1: Panel c:

- What does the 's' mean in s/10yr? Please explain in the caption.

Added: "...The s/10 yr indicates the slope per decade"

- What are the black dots on the figure added since 1985? Please add to the caption.

Modified from:

"...The left y-axis shows simulated meltwater (bars), and the right y-axis indicates the total area affected by extreme melting events each summer (**points**), derived from passive microwave satellite observations..."

We have added:

“...The left y-axis shows simulated meltwater (bars), and the right y-axis indicates the total area affected by extreme melting events each summer (**black points**), derived from passive microwave satellite observations...”

- What are the black lines on the figure? It looks like they are the decadal trends. If so, say so in the caption.

Added: “...The solid black lines represent the regression lines for total summer meltwater accumulation (1950–2023 and 1990–2023) and total area (1985–2023)”

- Since you are comparing the time spans 1950-1975 and 2000-2023 in panel a, it would be informative to have these slopes per 10yr as well.

In this case, we prefer to retain the current information. We believe that adding more details—there are already three labels in Figure 1c—would make it difficult to read.

Figure 2: Panel b: why start the time series in 1980 when your analysis starts with year 1950?

This is because the figure zooms in on the seven most extreme melt events, all of which occurred after 1980, making this period the most relevant for analysis. Consequently, Panel b of Figure 2 focuses on the changes since 1980.

Figure S2: The subpanel associated with CWT 13 shows a decreasing red line meaning the slope/10yr should be negative and read -20.959.

Corrected.

Table S1: the total meltwater for year 1997 looks more like 25 Gt (based on Figure 1).

We believe this comment arises from a misunderstanding of the metrics used to define extreme melting events and of the data presented in the figures.

To clarify, Figure 1b shows the daily peak meltwater associated with each event, not the seasonal or total summer meltwater. For the August 1997 event, the daily peak meltwater reaches 16.3 Gt (without detrending) and 11.6 Gt (detrended, as shown in Figure 1b). Table S1 reports these peak daily values as well as the accumulated meltwater during each event, which amounts to 52 Gt (without detrending) and 29.1 Gt (detrended) for 1997. The accumulated values per event are not shown in Figure 1 and are provided only in Table S1 as complementary information on event magnitude.

We confirm that 1957 exhibits a total summer meltwater amount exceeding 250 Gt; however, this is not exceptional, as more than 90% of melt seasons reach similar values (Figure 1c). In 1957, summer melt occurred without any individual daily peak meltwater event ranking among the top 10 extremes. For this reason, 1957 does not appear in Figure 1b or Table S1, which are based exclusively on peak daily meltwater intensity rather than total seasonal melt. Regarding the potential typo in Table S1, we have rechecked the values and confirm that the reported numbers for 1997 are correct.

Reviewer #1 (Remarks to the Author):

I thank the authors for thoroughly addressing my previous review. I have just a couple of remaining minor points.

We would like to thank the reviewer for their evaluation of our work.

L3: There should be a space between the end of the title and the lead author.

Corrected.

L16: I think 'of' would be better than 'in'.

Corrected.

L50: if these factors enhanced the rise, it might be worth stating briefly what caused the underlying rise.

Changed from:

“These circulation-induced factors enhanced a rise in extreme runoff recurrence in GrIS”

To:

“Increasing anticyclonic and blocking events enhanced surface melting, contributing to the rise in extreme runoff recurrence in GrIS”

L224: It seems to be Figure 1c that shows the temporal trend. Also, I think it would be worth stating clearly (in the figure caption, maybe) that in this case 'area' is the sum of the area experiencing melting each day per year (I think). Otherwise it could be misleading that the decadal trend (2.8M km²) in melt area is far greater than the total area of the ice sheet (~1.8M km²).

Corrected. Each pixel contributes for every day it experiences melt, so the seasonal total may exceed the physical area of the GrIS.

Added a figure note: “... total area, calculated by the cumulative sum of daily meltwater pixels affected by extreme melting events each season ...”

Reviewer #2 (Remarks to the Author):

I am happy with the revisions.

Reviewer #3 (Remarks to the Author):

I would like to thank the authors for their careful considerations in replying to all the reviewers comments. I personally have no other comments on this study. Once the comments from the other reviewers are addressed, I believe this work would be ready for publication.